Transcriptome analysis of the differences between two kinds of Cassia nomame germplasm resources

Li Jin 1
Xu Ningwei 2
Xu Xingyou 3
Bai Zhiying 1 zhiyingbai@126.com
1 College of Life Sciences, Hebei Agricultural University , Baoding , China
2 School of Urban Construction, Beijing City University , Beijing , China
3 College of Marine Resources & Environment, Hebei Normal University of Science & Technology , Qinhuangdao , China
Abd El-Moneim Diaa
Electronic publication date: 2025 Oct 30
Publication date: 2025
Volume: 13
Electronic Location ID: e20261
Received 2025 May 19; Accepted 2025 Sep 29
Copyright: © 2025 Li et al.
Copyright year: 2025
Copyright holder: Li et al.
License: This is an open access article distributed under the terms of the Creative Commons Attribution License, which permits unrestricted use, distribution, reproduction and adaptation in any medium and for any purpose provided that it is properly attributed. For attribution, the original author(s), title, publication source (PeerJ) and either DOI or URL of the article must be cited.
License URL: https://creativecommons.org/licenses/by/4.0/

Keywords: Cassia nomame, De novo transcriptome, Secondary metabolism, Transcription factor

Funding: State Key Laboratory of North China Crop Improvement and Regulation NCCIR2020ZZ-18 Natural Fund Project of Hebei Province C2022204036 This study was supported by the State Key Laboratory of North China Crop Improvement and Regulation (NCCIR2020ZZ-18), Natural Fund Project of Hebei Province (C2022204036). The funders had no role in study design, data collection and analysis, decision to publish, or preparation of the manuscript.

==============================
Cassia nomame belongs to the genus Cassia of the leguminous Cassia subfamily. It is used as a traditional wild Chinese herbal medicine with a long history and rich medicinal use. However, owing to limited germplasm resources and research methods, there is still a lack of understanding of C. nomame at the molecular level, especially the differences between various cultivars and secondary metabolic regulatory genes. In this study, we performed de novo transcriptome assembly of two C. nomame cultivars with different characteristics using transcriptome method. A total of 56,136 unigenes were obtained, of which 34,783 genes were annotated, including 7,309 candidate transcription factors (TFs) of 57 TF families. Through differential expression analysis, we identified 4,696 differentially expressed genes (DEGs). The results of Gene Ontology (GO) and KEGG functional enrichment analysis revealed that the DEGs were mainly involved in secondary metabolite biosynthetic process, transcriptional regulation, response to hormone, growth, and development. TF family analysis and verification experiments showed that these TFs were significantly different expressed in the two C. nomame germplasm resources, which suggested that they might be important genes affecting the traits of C. nomame. In conclusion, the results of this study are significant for mining C. nomame germplasm resources and enhancing our understanding of the formation of different germplasm resources and medicinal ingredients mining.

Introduction

Cassia nomame (Sieb.) Kitagawa belongs to the genus Cassia of the family Leguminosae. It is a 1 year old herb mainly distributed in China, Korea, Japan, India, and other countries (Sun et al., 2022), and mostly grows as shrubs or grasses in mountainous and open areas. It generally has 8 to 28 pairs of leaves, with black-brown, disc-shaped sessile glands at the upper end of the petiole. The leaflets are lanceolate, the peanuts exist in the axils of the leaves and are yellow in color, and the pods are flat. C. nomame, as a traditional wild Chinese herbal medicine with a long history, has a variety of medicinal applications. It is mainly used for the treatment of edema, chronic constipation, cough, phlegm, and nephritis, and can be employed for deworming. Besides, C. nomame is also used as a drink and dish with good taste because of its rich nutritional value (Heo et al., 2023).

Currently, research on C. nomame mainly focuses on the functional components and pharmacology, and studies have found that the aerial parts and seeds of C. nomame exhibit medicinal activity. The main functional chemical components of C. nomame include flavonoids and their glycosides, anthraquinones and their glycosides, flavanols and their glycosides, etc. (Liao et al., 2020). In addition to anthraquinones isolated from the seeds and aerial parts of C. nomame, flavanoids and other flavonoids, including luteolin, luteolin 7-glucoside, vitexin, 7-isocassiaoccidentalin B, demethyltorosaflavones C and D, etc., have also been detected (Kitanaka & Takido, 1992; Syed et al., 2016). Furthermore, chrysophanol, emodin monomethyl ether, and emodin have been isolated from the aerial parts, and a series of emodin and anthrone have been extracted from the seeds of C. nomame (Kitanaka & Takido, 1992). In-depth analysis of the seed components of C. nomame showed the presence of one flavonoid and six flavanols, among which five compounds had inhibitory effects on lipase (Hatano et al., 1997; Marchitto et al., 2018).

Many recent studies have shown that C. nomame can not only lower blood lipids, reduce blood pressure, protect liver, and act as laxatives, but can also enhance immunity, bacteriostasis, and antioxidation, as well as protect the nervous system. It has been reported that chromone derivatives extracted from C. nomame can inhibit the activity of tobacco mosaic virus (Liao et al., 2020). Intragastric administration of the aqueous extract of C. nomame has been noted to significantly increase the activity of lactate dehydrogenase in the ciliary muscle and adenosine triphosphate content in the eye tissue, expand the peripheral blood vessels, and improve blood circulation of retina and nerve (Marchitto et al., 2018). Studies have shown that C. nomame ethanol extract (CSE) exerted a protective effect on acute liver injury induced by CCl4 in rats. CSE has been noted to reduce the malondialdehyde content in rat liver and mitochondria, and enhance the activity of superoxide dismutase, glutathione reductase, and glutathione-S-transferase in rat serum and liver mitochondria, indicating that CSE can scavenge free radicals in vivo and play a role in protecting the structure and function of cell membrane (Lim & Lee, 2012; Heo et al., 2023). Furthermore, C. nomame extract has been reported to inhibit chromosomal aberrations and protect cells from damage (Konishi et al., 2004). In addition, C. nomame crude extract has been found to inhibit apoptosis and reduce brain injury in rat ischemia-reperfusion model, thus providing a novel strategy for C. nomame application (Kim & Lee, 2010).

Currently, research on C. nomame has intensified around the world, and despite considerable progress, there are still various unexplored issues. For example, the specific pharmacological mechanism, best extraction process, and safety of long-term use of C. nomame need further investigated. Nevertheless, continuous advancement in science and technology can provide a deeper understanding of this remarkable plant and offer wider insights into its application in medicine, healthcare, and other fields.

Materials and Methods

Plant materials

C. nomame (T4-1 and T4-2) grows in Xinglong and Kuanchen respectively, Chengde City, Hebei Province, China. C. nomame (T4-1, T4-2) leaves of flowering stage with three replicates were collected and gathered, promptly frozen in liquid nitrogen bottles, and cryopreserved to extract RNA and metabolites. Subsequently, the leaves were placed in an ultra-low-temperature refrigerator.

RNA extraction, library construction, and sequencing

Total RNA was extracted from the samples using TRIzol (Invitrogen, Waltham, MA, USA) and purified utilizing RNeasy column (Qiagen, Hilden, Germany). PolyA-tailed mRNA was isolated from total RNA using oligo(dT) magnetic beads. The purified RNA was then fragmented into approximately 300 bp pieces via ion-mediated shearing. First-strand cDNA was generated from the RNA fragments with random hexamer primers and reverse transcriptase. Subsequently, the second-strand cDNA was produced using the first strand as the template. After library construction, the library fragments were enriched by PCR amplification, and the library was selected according to the fragment size. The library size was 450 bp. The quality of the library was verified by Agilent 2100 Bioanalyzer, and the total and effective concentrations of the library were determined. Based on the library’s effective concentration and the required sequencing depth, pools were created by combining uniquely indexed libraries in specific ratios. This pooled library was diluted uniformly to a 2 nM concentration and then denatured with alkali to generate single-stranded DNA. The resulting libraries were subjected to paired-end sequencing on an Illumina platform.

Transcriptome data filtering and assembly

Data filtering using Cutadapt was employed to remove the 3′-end joint sequence, and reads with an average mass fraction lower than Q20 were eliminated. For transcriptome sequencing without reference genome, Trinity software was used to splice clean reads to obtain transcripts for subsequent analysis (Haas et al., 2013). Trinity is a de novo assembly software for transcriptome assembly, which is based on the principle of De Bruijn Graph (DBG) assembly to splice high-quality sequences. The software consists of three independent software modules with the following workflow: Inchworm: a short-sequence library of K-mer length is constructed using high-quality sequences, and the short sequence is extended by overlap of K-mer-1 length between short sequences to obtain a preliminary spliced contig sequence; Chrysalis: the contig sequences are clustered based on overlap between them, and a Bruijn graph is constructed for each class; and Butterfly: deals with these Bruijn diagrams, and finds the path according to the reads and paired reads in the diagram to obtain the transcript. After completion of stitching, a transcript sequence file in FASTA format can be obtained. The longest transcript under each gene is extracted as the representative sequence of the gene, called unigene.

Transcriptome functional annotation and differential gene expression analysis

The databases used for gene function annotation included NR (NCBI non-redundant protein sequences), Gene Ontology (GO), Kyoto Encyclopedia of Genes and Genome (KEGG), Evolutionary genealogy of genes: Non-supervised Orthologous Groups (eggNOG), Pfam, and SwissProt. By comparing and annotating with the NR library, the similarity between the gene sequences of the target species and related species as well as the functional information of the gene of the target species can be obtained. GO annotation was completed using BLAST2GO software, and the default parameters of BLAST2GO were used for annotation (Conesa & Gotz, 2008). The GO annotation results were mapped to GO terms, and the number of genes annotated to the second-level classification was counted. KO and Pathway annotations were mainly performed using the KOBAS annotation system. After completing the KO annotation, the KO was mapped to the corresponding KEGG pathway. The eggNOG comparison annotation was performed on the gene, and the eggNOG number of the best comparison result was assigned to the corresponding gene (Cantalapiedra et al., 2021). Furthermore, each gene was classified into the eggNOG classification directory by using the correspondence between the eggNOG number and eggNOG classification directory.

Transcriptome expression quantification was performed using the RSEM software. With the transcript sequence as a reference, the clean reads of each sample were compared to the reference sequence (Li & Dewey, 2011). Then, the number of reads on each gene was calculated for each sample, and the FPKM value for each gene was computed. We used DESeq2 to analyze the difference in gene expression (Love, Huber & Anders, 2014), and the conditions for screening differentially expressed genes (DEGs) were as follows: | log2FoldChange | > 1, FDR < 0.05. In total, six samples were included and are evenly separated into two groups, with three biological replicates on each group. The statistical power of this experimental design, calculated in RNASeqPower is 0.85 (FDR < 0.05) (Hart et al., 2013).

Functional and pathway enrichment analysis

GO term enrichment analysis was conducted with the topGO package. Statistically significant terms were identified by applying a hypergeometric test to the differentially expressed genes (DEGs) annotated under each term, using the whole genome as background. A false discovery rate (FDR) threshold of <0.05 was applied to define significantly enriched terms. The results were categorized into biological process (BP), molecular function (MF), and cellular component (CC). For each category, the ten most significant terms (lowest P-value) were selected for visualization. The KEGG database systematically analyzes gene functions and links genomic information and functional information (Kanehisa et al., 2023). The KEGG metabolic pathway enrichment analysis was performed using KOBAS online software, and KEGG pathway with FDR < 0.05 was selected as the enrichment result (Bu et al., 2021).

Transcription factor analysis and real-time quantitative PCR

The process of eukaryotic transcription initiation is very complex and often requires the assistance of a variety of protein factors. The transcription factors (TFs) are a class of protein molecules that can specifically bind to specific sequences upstream of the 5’ end of the gene and form a transcription initiation complex with RNA polymerase II to participate in the process of transcription initiation. The transcription factor (TF) prediction was compared with PlantTFDB (Plant Transcription Factor Database) database to predict the TFs and obtain the family information of the TFs (Jin et al., 2017). The DEGs predicted as TFs were statistically analyzed, and according to the family information of the TFs, the number of differentially expressed TFs in each TF family in the comparison group was statistically analyzed.

Using the Total RNA Purification Kit (Promega, Madison, WI, USA), the samples’ total RNA was extracted. DNase I was then used to degrade the RNA and the RevertAid First Strand cDNA Synthesis Kit (Thermo Fisher Scientific, Waltham, MA, USA) was utilized to synthesize cDNA. Three biological replicates of qRT-PCR were run using the SYBR green PCR Master Mix. For equivalent loading, the 18S RNA was utilized as an internal standard. The 2−ΔΔCt analysis method was used to determine the relative expression levels. The SD of three biological replicates was calculated using GraphPad Prism 8 software, and the student’s t-test analysis was performed. List of primers utilized in this investigation was listed in Table S1.

Protein-protein interaction network analysis

Search Tool for the Retrieval of Interacting Genes/Proteins (STRING) is a protein–protein interaction (PPI) database developed by EMBL, and includes the most powerful experimental evidence, data mining, and homologous prediction of PPIs (Szklarczyk et al., 2023). By including the PPI information of the species in the STRING database and based on the results of DEGs analysis, we screened PPI pairs with DEGs as both ends of the nodes and score > 0.95 in the direct database. When there was no PPI information for the species in the STRING database, we selected similar species to compare the sequences and then obtained the relationship between the proteins of the species.

Yeast two-hybrid assay

The full-length coding sequences of DN82231_c0_g1 and DN55_c6_g1 were amplified by PCR and combined with pGADT7 and pGBKT7 vectors, respectively. GAL4 two-line hybrid system was used for Y2H detection assay and the recombinant plasmid was transformed into Y2H golden yeast strain. The transformed yeast cells were cultured at 30 °C for 3 days in selective medium lacking leucine, tryptophan and selective medium lacking leucine, tryptophan, histidine and adenine, respectively. Then, the growth of yeast cells was observed and reflected the interaction between the two proteins.

Results

Transcriptome data assembly and annotation

To explore the differences among various C. nomame germplasm resources, we selected two C. nomame varieties with significant differences in growth and development, stress response and secondary metabolites based on previous experiments (Fig. S1). Then, we performed de novo transcriptome sequencing for the leaves of two C. nomame cultivars which have different growth status and anthraquinone content (Figs. 1A and S1). Based on the transcriptome data after quality control and screening, the clean reads of the six samples were in the range of 5,747,748,258–6,985,352,110 bp, the percentage of Q30 bases was 93.09–93.83%, and the GC content was 40.17% (Tables 1 and S2). The overall sequencing data filtering quality was good, and could be used for subsequent transcriptome analysis. Through transcriptome assembly, we obtained 56,136 independent genes, with length ranging from 302 to 16,566 bp, average length of 1,088 bp (Fig. 1B), and total length of 66,667,714 bp (Table 1). According to the distribution of different lengths of independent genes after assembly, the gene length of 0–500 bp accounted for about 60% of the total length. Furthermore, the length of N50 was 2,096 bp, which indicated that the assembly results were reliable.

Figure 1 Statistical analysis of the de novo transcriptome data of C. nomame.

(A) Two different cultivars of C. nomame. Scale bar indicates 10 cm. (B) Length distribution of unigenes in the transcriptome assembly of C. nomame. (C) Upset plot of unigenes annotation in the NR, KEGG, GO, Pfam, eggNOG, and SwissProt databases.

Table 1 Overall statistics of transcriptome sequencing sequence of Cassia nomame.

	Transcript	Unigene	
Total length (bp)	276,671,752	66,667,714	
Sequence number	163,874	56,136	
Max length (bp)	16,566	16,566	
Mean length (bp)	1,688.32	1,187.61	
N50 (bp)	2,453	2,096	
N90 (bp)	825	451	
GC%	40.23	40.17	

To explore the functions of unigenes, we used five databases (NR, KEGG, GO, eggNOG, SwissProt) to annotate the obtained unigenes (Fig. 1C). A total of 34,783 unigenes with annotation information were obtained, 21,353 unigenes were not annotated, and the annotation ratio was 62%. Among them, 32,444, 19,625, 12,380, 30,361, 20,215, and 26,647 unigenes were annotated in the NR, GO, KEGG, eggNOG, Pfam, and SwissProt databases, respectively, with 7,490 unigenes annotated in all the five databases.

Analysis of DEGs in transcriptome

To ensure accuracy of subsequent analysis, we first corrected the sequencing depth and then adjusted the length of the gene or transcript to obtain the RPKM value of the gene. The expression distribution was calculated according to the RPKM value of each gene, which revealed relatively uniform gene expression among different samples, indicating that these results can be used for subsequent DEGs analysis (Fig. 2A). Principal component analysis (PCA) and Pearson correlation coefficient analysis were employed to cluster the samples according to the expression level. The PCA results showed that the three biological replicates of different C. nomame cultivars were distributed in the same region, indicating data integrity and reliability, as well as diverse expression levels of different genes in the two cultivars. Pearson correlation coefficient analysis revealed that the correlation between the two samples was >0.99 (Fig. S2), implying that the samples had good repeatability. Based on the gene expression level in the transcriptome data, we selected genes with differential expression fold | log2FoldChange | > 1 and P < 0.05 as DEGs. Finally, a total of 4,696 DEGs were identified between the two C. nomame data, of which 2,391 and 2,305 DEGs were upregulated and downregulated, respectively (Figs. 2C and S3).

Figure 2 Unigenes expression analysis of the two C. nomame cultivars.

(A) Expression level analysis of different biological replicates of C. nomame T4-1 and T4-2. (B) PCA of the expression datasets of C. nomame T4-1 and T4-2. (C) Volcano plots of DEGs in C. nomame T4-1 and T4-2. Genes with expression fold change >2 and P < 0.05 were filtered as differentially expressed.

Functional analysis of DEGs

GO is an internationally standardized classification of gene functions, which provides a biological basis for the global characterization of de novo assembled transcripts (Gene Ontology Consortium, 2015). To elucidate the biological function of DEGs in C. nomame, we performed GO functional enrichment analysis of selected 4,696 DEGs (FDR ≤ 0.05), which included three aspects of biology, namely, BP, MF, and CC. The results revealed that the DEGs of the two C. nomame cultivars were mainly enriched in BP. In the BP classification, the GO terms were mainly enriched in the regulation of DNA binding, secondary metabolites, photosynthesis redox, cell division response, pigment biosynthesis, polysaccharide metabolism, hemicellulose metabolism, response to plant hormones, tetrapyrrole biosynthesis process, flavonoid and terpenoid anabolism, gibberellin-mediated signaling pathway, phylloquinone biosynthetic process, quinone biosynthetic process, and other biological processes (Fig. 3 and Table S3). In the MF classification, the GO terms were significantly enriched in TF activity, tetrapyrrole binding, pigment binding, quinone binding, terpenoid synthase activity, gibberellin oxidase activity, sequence-specific DNA binding, rRNA binding, and dioxygenase activity (Fig. 4 and Table S3). In the CC classification, the GO terms were mainly enriched in chloroplast thylakoid, thylakoid photosystem, cell wall plastid, ribosome, photosystem I reaction center, plant cell wall, ATPase complex, etc. (Fig. 5 and Table S3). These results suggested that the differences in biological functions may lead to variations in the growth status and secondary metabolites of different C. nomame cultivars.

Figure 3 Biological progress enrichment analysis of DEGs in C. nomame T4-1 and T4-2.

Fisher’s exact test and adjusted P-value calculation were performed by Benjamini-Yekutieli method (FDR < 0.05).

Figure 4 Molecular function enrichment analysis of DEGs in C. nomame T4-1 and T4-2.

Fisher’s exact test and adjusted P-value calculation were performed by Benjamini-Yekutieli method (FDR < 0.05).

Figure 5 Cellular component enrichment analysis of DEGs in C. nomame T4-1 and T4-2.

Fisher’s exact test and adjusted P-value calculation were performed by Benjamini-Yekutieli method (FDR < 0.05).

KEGG annotation provides information of transcripts related to metabolic process and functions in cellular processes (Kanehisa et al., 2023). We used the KEGG metabolic pathway enrichment analysis to determine the key metabolic pathways of the DEGs of the two C. nomame cultivars. The results revealed that the DEGs of the two C. nomame cultivars were significantly enriched in ribosome, photosynthesis, flavonoid biosynthesis, carotenoid biosynthesis, phenylpropanoid biosynthesis, anthocyanin biosynthesis, nitrogen metabolism, circadian rhythm, α-linolenic acid metabolism, tryptophan metabolism, terpenoid biosynthesis, plant hormone signal transduction, and other metabolic pathways (Fig. 6 and Table S3). KEGG functional enrichment analysis showed that the DEGs played an important role in the biosynthesis and metabolism of secondary metabolites, transcriptional regulation, and other processes.

Figure 6 Function annotation of the DEGs of C. nomame T4-1 and T4-2 based on KEGG classification (FDR < 0.05).

The abscissa indicates the degree of significant enrichment.

Transcription factor identification and differential expression analysis

In eukaryotes, TFs are proteins that bind to sequence-specific DNA. They control gene expression by binding to specific DNA sequences to ensure that they are expressed in the appropriate number of cells at the right time throughout the life cycle of cells and organisms (Lambert et al., 2018). TFs are mainly involved in the initial stage of RNA transcription and are the key factors in regulating the gene expression levels (Strader, Weijers & Wagner, 2022). GO enrichment analysis showed that transcription regulator activity, regulation of DNA binding, and other GO terms related to TFs were significantly enriched, indicating that the regulation of TFs in the two C. nomame cultivars played an important role in the formation of their phenotypes. Furthermore, we predicted the TFs of all unigenes and identified 7,309 candidate TFs. The global TF classification noted that the identified candidate TFs belonged to 57 TF families, such as bHLH, MYB, NAC, ERF, C2H2, MYB-related, WRKY, bZIP, C3H, etc. (Fig. 7A).

Figure 7 Transcription factor identification in de novo transcriptome data of C. nomame.

(A) Statistics of transcription factor family in C. nomame. (B) Differentially expressed transcription factors in C. nomame T4-1 and T4-2, respectively.

In addition, we also performed differential expression analysis of TF families, and found that 48 TF families were upregulated or downregulated in the two C. nomame cultivars. Some key regulatory gene families involved in response to abiotic and biotic stresses, such as WRKY, bHLH, MYB, NAC, bZIP, C2H2, HSF, and C2C2-YABBY, were upregulated or downregulated in both the cultivars (Fig. 7B). However, several TF family genes, such as ERF, C2H2, NF-YA, and TALE, were upregulated in T4-2, while TF families such as GRAS, C3H, BBR-BPC, and STAT were highly upregulated in T4-1, indicating that these TFs may be highly significant for the generation of various characteristics of C. nomame. To verify the authenticity of the transcriptome data, we used qRT-PCR technology to confirm the expression levels of individual genes in some TF families (Fig. 8). The results showed that the expression trend of these genes was consistent with RNA-Seq, which further validated our speculation.

Figure 8 RT-qPCR analysis of differentially expressed transcription factors in C. nomame T4-1 and T4-2.

Data represent mean ± SD of three independent biological replicates, and the asterisks indicate significant differences as assessed by Student’s t-tests (**P < 0.01).

Identification and verification of PPI network

Proteins are the main catalysts, structural elements, signal transmitters, and molecular machines of biological tissues. PPI is very important in the coordination of intracellular events, and is the basis of multiple signal transduction pathways in cells and various transcriptional regulatory networks (Von Mering et al., 2002). To describe the relationship between proteins encoded by the DEGs of the two C. nomame cultivars, we used STRING software to construct a PPI network to evaluate the relationship between protein-encoded DEGs. Based on the results of differential gene expression analysis, we screened PPI pairs with DEGs at both the ends and score > 0.95 in the direct database, and identified 675 interactions between proteins encoded by 171 DEGs (Fig. 9A). Most of these interacting proteins were noted to be encoded by upregulated genes, indicating that these proteins may play an important role in the growth and development of C. nomame. Subsequently, we randomly selected a pair of interacting proteins DN82231_c0_g1 (Malectin domain kinesin) and DN55_c6_g1(Gamma subunit of Arabidopsis chloroplast ATP synthase), and verified them by yeast two-hybrid technology. The findings revealed an interaction between the two proteins (Fig. 9B), which further confirmed the authenticity of the interaction network.

Figure 9 PPI network of the DEGs. (A) PPI network constructed using 171 DEGs and 675 edges.

Node size indicates node degree, node color is based on DEGs. (B) Verification of PPI between DN82231_c0_g1 and DN55_c6_g1 based on Y2H method.

Discussion

C. nomame is a wild plant with many important applications such as medicine, healthcare, and food, and has significant potential for economic development (Konishi et al., 2004). The lack of resource protection awareness and standardized cultivation method has severely affected C. nomame wild resources, necessitating measures for mining and conservation of germplasm resources. C. nomame is a natural herb that contains a variety of bioactive compounds, such as flavonoids, ketones and others, which may help support healthy metabolism and promote weight loss (Hatano et al., 1997). Based on the biological characteristics of C. nomame seeds, the present study explored the differences in gene expression between different C. nomame cultivars and their roles in biological processes, and the results obtained provide an important theoretical basis for the utilization and development of C. nomame germplasm resources. We assembled the unigenes of the two C. nomame cultivars by de novo assembly of transcriptome data, and successfully obtained 56,136 unigenes and 163,874 transcript information. Among the obtained unigenes, 34,783 unigenes had at least one database annotation information, with an annotation ratio of 62%, thus achieving sufficient sequence coverage depth and acceptable assembly results as well as expression level detection. The de novo assembly of transcriptome sequencing can improve the genome annotation of C. nomame, and the specificity of transcript signal can help to distinguish individual members of different gene families.

The results of the present study showed that investigation of plant metabolites is crucial for identification of metabolites, differentiation of cultivars, and molecular breeding. Traditional Chinese medicine uses natural products, with secondary metabolites being the major active ingredients, which are inherently unstable. Besides, factors such as plant resources, origin of medicinal materials, processing technology, and storage conditions can vary, and the internal quality of the same type of medicinal materials may be difficult to stabilize. The chemical constituents of C. nomame are complex and have a wide range of pharmacological effects and clinical applications. Modern pharmacological studies have determined two main categories of effective components of C. nomame, namely, anthraquinone derivatives (which have anticancer, antibacterial, and diuretic effects) and anthraquinone glycosides (which have effects on lowering blood lipids, reducing blood pressure, detoxification, and liver protection). In the present study, GO terms related to quinones anabolism, such as phylloquinone biosynthetic process, quinone biosynthetic process, quinone binding, oxidoreductase activity, and quinone or similar compound as acceptor, were significantly enriched with DEGs, indicating that the expression of genes related to the synthesis and metabolism of quinones varies in the two C. nomame cultivars. Analysis of the DEGs revealed that the related enriched genes were expressed at a higher level in the T4-2 cultivar with better growth status (Fig. S4), implying that C. nomame T4-2 may be rich in quinone compounds and might contain higher content of secondary metabolites such as anthraquinone derivatives.

Flavonoids are widely present in Chinese medicinal materials, and exhibit a variety of pharmacological effects such as antioxidant effects, scavenging free radicals in the body, and protecting cells from oxidative damage. As flavonoids are ubiquitous in C. nomame, the metabolites of this plant have gained widespread attention owing to their health advantages. Some studies have isolated flavonoids from the aerial parts of C. nomame, including nordihydroflavone C and D, 7-hydroxychromone, luteolin, vitexin, and luteolin-7-glucoside (Syed et al., 2016), and their structures have been verified based on spectral evidence (Kitanaka & Takido, 1992). Luteoloside (Luteolin-7-O-glucoside) is a natural flavonoid widely found in C. nomame, honeysuckle, and other plants, and is one of the main active compound in C. nomame. It helps in reducing body weight, decreasing lipid weight, regulating blood lipids, and reducing obesity (Fan et al., 2014). Modern pharmacological studies have shown that luteoloside has antimicrobial, antibacterial, anticancer, and other beneficial effects (Li et al., 2019). In the present study, GO terms related to flavonoid anabolism, such as flavonoid metabolic process, flavonoid biosynthetic process, flavonol metabolic process and others, were significantly enriched with DEGs, indicating that genes related to flavonoid anabolism were differentially expressed in the two C. nomame cultivars. In addition, the biosynthetic metabolic processes of terpenoids, carotenoids, phenylpropanoids, phytosteroids, and other compounds were also significantly enriched with DEGs in the two cultivars, indicating that the metabolites of C. nomame cultivars considerably vary.

Transcription is a dynamic process, and TFs are essential for the regulation of gene expression, especially for regulating genes involved in the synthesis and metabolism of secondary metabolites in plants. Flavonoids biosynthesis is regulated by a variety of TFs, and R2R3-MYB, bHLH, and WD40 repeat proteins have been reported to be essential for the synthesis and metabolism of flavonoids (Hichri et al., 2011). Studies have shown that MYB TFs can regulate the synthesis of flavonols by activating early biosynthetic genes PAL, C4H, CHS, CHI, and F30H (Pandey, Misra & Trivedi, 2015). It must be noted that bHLH is a key TF in the regulation of flavonoid biosynthesis, and activates the secondary metabolite synthesis genes by forming a highly dynamic MYB-bHLH-WD40 complex, such as F30H, ANR, DFR, and UFGT (Pireyre & Burow, 2015; Xu, Dubos & Lepiniec, 2015). In Arabidopsis, bHLH TF has been found to regulate the biosynthesis of anthocyanins and proanthocyanidins by directly or indirectly interacting with MYB transcription factor (Chen et al., 2023). TFs related to flavonoid biosynthesis, including MYB, bHLH, and WD40, have been determined to be differentially expressed during leaf senescence in Lonicera macranthoides, and MYB12, MYB75, bHLH113, and TTG1 are considered to be the key factors involved in regulating luteolin biosynthesis (Chen et al., 2018). Furthermore, TFs such as AP2, ERF, C2H2, and C3H have been shown to play an important role in the regulatory network of plant secondary metabolites (Zhou & Memelink, 2016; Kong et al., 2022; Han et al., 2023). These findings reveal that TFs play a vital role in the regulation of secondary metabolites synthesis and metabolism in plants, are key regulators that can activate or inhibit the biosynthesis of natural plant products, and can effectively enhance the synthesis of required secondary metabolites.

In the present study, the expression levels of MYB, bHLH, ERF, C2H2, and other TF families related to plant secondary metabolism were found to significantly vary, indicating that the two C. nomame cultivars examined have different metabolic regulation processes. Besides, qRT-PCR assay to verify the randomly selected differentially expressed TFs revealed that their expression levels were mostly consistent with the RNA-Seq data, which further confirmed the reliability of data and possibility that these genes may play different functions in the two C. nomame cultivars. In the follow-up experiments, the study of these differentially expressed TFs has important guiding significance for changing the characteristics of C. nomame varieties and regulating the production of secondary metabolites. In addition, functional analysis of DEGs demonstrated that the GO and KEGG terms related to plant hormone response and growth and development, such as response to hormone, gibberellin-mediated signaling pathway, response to cytokinin, cell wall organization or biogenesis, and regulation of seedling development, were significantly enriched. Gibberellin can promote cell division and elongation, accelerate the growth of roots, stems, and leaves, and promote flower bud differentiation and flowering during plant growth. Besides, it can also promote photosynthesis and increase the accumulation of photosynthetic products in plants (Davière & Achard, 2013). In the present study, the growth status of C. nomame T4-2 was significantly better than that of C. nomame T4-1, indicating that the DEGs related to gibberellin, cytokinin, growth, and development are crucial for the formation of C. nomame T4-2 germplasm resources (Fig. 1A).

Protein-protein interaction (PPI) networks consist of interacting proteins that collaboratively regulate fundamental biological processes, including signal transduction, gene expression, energy and metabolite metabolism, and cell cycle control. Systematically investigating large-scale protein interactions within biological systems is crucial for deciphering the functional mechanisms of proteins, elucidating the principles of signal response and energy metabolism under pathological or other specific physiological conditions, and revealing functional associations among proteins—often in conjunction with the identification of key genes. In this study, we performed PPI analysis on differentially expressed genes, and identified 675 interactions between proteins encoded by 171 DEGs. The interaction between DN82231_c0_g1and DN55_c6_g1 was confirmed by testing a pair of interacting proteins in the interaction network, which further proved the reliability of the protein interaction network. DN82231_c0_g1 and DN55_c6_g1 were Malectin domain kinesin and Gamma subunit of Arabidopsis chloroplast ATP synthase, respectively. They were differentially expressed and interact with each other in two C. nomame, indicating that these two proteins played an important role in regulating the important traits of C. nomame. These results also laid an important theoretical foundation for further exploration of important genes regulating important traits of C. nomame.

Conclusions

Our study attempted to provide an important overview of the transcriptome analysis of C. nomame by sequencing the RNA of two C. nomame germplasm resources. The results offer an in-depth understanding of the gene expression pattern in C. nomame and provide some interesting insights for further research. The transcriptome assembly and transcriptome data obtained for the two C. nomame cultivars could be crucial for the genome-wide annotation of C. nomame. Further analysis of the transcriptome data could provide a deeper understanding of the differences between the two C. nomame germplasm resources, which could be significantly important for the protection of C. nomame germplasm resources and mining of medicinal values.

Supplemental Information

Supplemental Information 1 The germination rate under drought stress treatment, and the content of secondary metabolites anthraquinone of C. nomame T4-1 and T4-2.

Supplemental Information 2 Correlation between the de novo transcriptome data of C. nomame T4-1 and T4-2.

Supplemental Information 3 Heatmap of DEGs in C. nomame T4-1 and T4-2.

Based on the Z-score strategy, the expression level is represented by values ranging from −2 to 2, which indicates low to high.

Supplemental Information 4 Heatmap of DEGs related to secondary metabolites such as anthraquinone derivatives in C. nomame T4-1 and T4-2.

Based on the Z-score strategy, the expression level is represented by values ranging from −2 to 2, which indicates low to high

Supplemental Information 5 Primers used in this study.

Supplemental Information 6 RNA quality check and sequencing data quality statistics.

Supplemental Information 7 Detailed information of enrichment GO terms and KEGG terms.

Supplemental Information 8 Raw data for RT-qPCR.

Additional Information and Declarations

Competing Interests

The authors declare that they have no competing interests.

Author Contributions

Jin Li conceived and designed the experiments, performed the experiments, analyzed the data, prepared figures and/or tables, authored or reviewed drafts of the article, and approved the final draft.

Ningwei Xu performed the experiments, prepared figures and/or tables, and approved the final draft.

Xingyou Xu performed the experiments, prepared figures and/or tables, and approved the final draft.

Zhiying Bai conceived and designed the experiments, authored or reviewed drafts of the article, and approved the final draft.

Data Availability

The following information was supplied regarding data availability:

The transcriptome data are available in National Genomics Data Center (NGDC) database: PRJCA024218.

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
