# Peer review of "Transcriptome analysis of the differences between two kinds of Cassia nomame germplasm resources"

_PeerJ, doi:10.7717/peerj.20261_

## Round 0.1 · original submission · Major Revisions

Dear Authors

The manuscript cannot be accepted for publication in its current form. The manuscript requires substantial revision to meet the journal's standards. The authors are invited to revise the paper in consideration of all suggestions made by all the reviewers, including the reviewer who rejected the manuscript. Please note that the requested changes are required for publication.

With Thanks

Reviewer 1 ·

Basic reporting

The rationale for comparing T4-1 and T4-2 is unclear. The manuscript does not clearly explain the significance of this comparison. If the goal is to examine differences in transcription factor (TF) expression between T4-1 and T4-2, both plants should be grown under identical controlled environmental conditions to avoid confounding variables. Conversely, if the intent is to assess the impact of different environmental conditions on TF expression, this objective needs to be clearly stated and justified in the manuscript.

Experimental design

RNA quality check, which also includes RNA integrity number, is missing. Quality assurance is very mandatory at each step to produce highly reliable data.

Validity of the findings

Figure 7b lacks clarity. The caption does not completely explain the content of the figure, and there is no reference to T4-1 and T4-2, which creates confusion about what is being depicted or compared.
The meaning of the coloured bars in Figure 8 is unclear, as no legend or explanation is provided. Additionally, the RT-PCR results are presented without any statistical analysis, which limits their interpretability. Furthermore, the selection of an appropriate housekeeping gene for normalisation is not addressed, which is critical for ensuring accurate gene expression analysis.
The list of primers in the supplementary material does not include the amplicon lengths.
The authors examined a potential protein-protein interaction between DN82231_c0_g1 and DN55_c6_g1 using the yeast two-hybrid (Y2H) assay; however, the rationale for focusing solely on these two genes is not clearly explained. Furthermore, the results and discussion section lack a detailed interpretation of the Y2H findings, limiting the reader's understanding of the assay outcome and its biological relevance.

Additional comments

The manuscript contains excessive explanatory content that detracts from the clarity and conciseness expected in scientific writing. The authors are encouraged to adopt a more precise and focused style to enhance the scientific rigour of the manuscript.

Reviewer 2 ·

Basic reporting

The manuscript is well written and gives sufficient evidence. However, data presentation is not satisfactory. The author requested to discuss the matter and solved the issue.

Experimental design

Statistical analysis was not done for example RT-qPCR analysis

Validity of the findings

sufficient

Additional comments

NA

Reviewer 3 ·

Basic reporting

The manuscript entitled “Transcriptome analysis of the differences between two kinds of Cassia nomame germplasm resources” focuses on the transcriptomic characteristics of C. nomame, a traditional wild Chinese medicine plant. However, the manuscript contains significant issues that require clarification and careful revision, and thus fails to meet the journal's basic standards at this stage.

Experimental design

One of the most important issues is that the authors proposed the two “varieties with significant differences in growth and development, stress response and secondary metabolites”. However, no details about these differences have been provided in the manuscript.
Figure 1A illustrates two different cultivars of C. nomame; however, the image provides no additional details beyond the plant size.

Validity of the findings

The presentation of experimental results has much content that is not standardized. For example, in lines 233-262, the authors suggest some GO
terms and KEGG metabolic pathways were significantly enriched. However, no corresponding GO term numbers or KO numbers were provided.
Similarly, corresponding transcripts numbers (the IDs of transcripts) would be expected after the gene names (e.g. lines 275-276, 280, 282-283).

Additional comments

The article also contains numerous syntax errors and terminology issues that need correction. Examples include “1-year-old herb” (line 38) and “has 8-28 pairs of leaves” (line 40).
The three panels in Figure 2 are too small, making the axis labels and other information difficult to distinguish.

---

## Round 0.2 · Minor Revisions

Dear Authors
The manuscript still needs a minor revision before publication. The authors are invited to revise the paper, considering all the suggestions made by the reviewer.
With Thanks

Reviewer 2 ·

Basic reporting

The manuscript is clear and understandable. Instead of repeatedly mentioning "transcription factors," the author can use TFs. The author can ensure that the abstract is written with quantified outcomes, rather than merely describing the overall result.

Experimental design

The number of technical and biological replicates should be provided for experiments.

Validity of the findings

Clear BUT in Fig. 1-6, the name of the parameter in the X axis is not mentioned, as I requested previously also.

Additional comments

n

Reviewer 3 ·

Basic reporting

no comment

Experimental design

no comment

Validity of the findings

no comment

Additional comments

no comment

---

## Round 0.3 · accepted · Accept

Dear Authors,

I am pleased to inform you that the manuscript has been improved following the last revision and can now be accepted for publication.

Congratulations on accepting your manuscript. Thank you for your interest in submitting your work to PeerJ.

With Thanks

Reviewer 2 ·

Basic reporting

The revised manuscript addressed all the issues. Therefore, the manuscript can be recommended for acceptance.

Experimental design

na

Validity of the findings

na

Additional comments

na